# AdaGAN: Boosting Generative Models

**Ilya Tolstikhin**
MPI for Intelligent Systems
Tübingen, Germany
`ilya@tue.mpg.de`

**Sylvain Gelly**
Google Brain
Zürich, Switzerland
`sylvaingelly@google.com`

**Olivier Bousquet**
Google Brain
Zürich, Switzerland
`obousquet@google.com`

**Carl-Johann Simon-Gabriel**
MPI for Intelligent Systems
Tübingen, Germany
`cjsimon@tue.mpg.de`

**Bernhard Schölkopf**
MPI for Intelligent Systems
Tübingen, Germany
`bs@tue.mpg.de`

## Abstract

Generative Adversarial Networks (GAN) are an effective method for training generative models of complex data such as natural images. However, they are notoriously hard to train and can suffer from the problem of *missing modes* where the model is not able to produce examples in certain regions of the space. We propose an iterative procedure, called *AdaGAN*, where at every step we add a new component into a mixture model by running a GAN algorithm on a *re-weighted* sample. This is inspired by *boosting* algorithms, where many potentially weak individual predictors are greedily aggregated to form a strong composite predictor. We prove analytically that such an incremental procedure leads to convergence to the true distribution in a finite number of steps if each step is optimal, and convergence at an exponential rate otherwise. We also illustrate experimentally that this procedure addresses the problem of missing modes.

## 1 Introduction

Imagine we have a large corpus, containing unlabeled pictures of animals, and our task is to build a generative probabilistic model of the data. We run a recently proposed algorithm and end up with a model which produces impressive pictures of cats and dogs, but not a single giraffe. A natural way to fix this would be to manually remove all cats and dogs from the training set and run the algorithm on the updated corpus. The algorithm would then have no choice but to produce new animals and, by iterating this process until there's only giraffes left in the training set, we would arrive at a model generating giraffes (assuming sufficient sample size). At the end, we aggregate the models obtained by building a mixture model. Unfortunately, the described meta-algorithm requires manual work for removing certain pictures from the *unlabeled* training set at every iteration.

Let us turn this into an automatic approach, and rather than including or excluding a picture, put continuous weights on them. To this end, we train a binary classifier to separate "true" pictures of the original corpus from the set of "synthetic" pictures generated by the mixture of *all the models* trained so far. We would expect the classifier to make *confident* predictions for the true pictures of animals missed by the model (giraffes), because there are no synthetic pictures nearby to be confused with them. By a similar argument, the classifier should make less confident predictions for the true pictures containing animals already generated by one of the trained models (cats and dogs). For each picture in the corpus, we can thus use the classifier's confidence to compute a weight which we use for that picture in the next iteration, to be performed on the re-weighted dataset.

The present work provides a principled way to perform this re-weighting, with theoretical guarantees showing that the resulting mixture models indeed approach the true data distribution.[1]

Before discussing how to build the mixture, let us consider the question of building a single generative model. A recent trend in modelling high dimensional data such as natural images is to use neural networks [1, 2]. One popular approach are *Generative Adversarial Networks* (GAN) [2], where the generator is trained adversarially against a classifier, which tries to differentiate the true from the generated data. While the original GAN algorithm often produces realistically looking data, several issues were reported in the literature, among which the *missing modes problem*, where the generator converges to only one or a few modes of the data distribution, thus not providing enough variability in the generated data. This seems to match the situation described earlier, which is why we will most often illustrate our algorithm with a GAN as the underlying base generator. We call it *AdaGAN*, for Adaptive GAN, but we could actually use any other generator: a Gaussian mixture model, a VAE [1], a WGAN [3], or even an unrolled [4] or mode-regularized GAN [5], which were both already specifically developed

---

**ALGORITHM 1** AdaGAN, a meta-algorithm to construct a "strong" mixture of $T$ individual generative models (f.ex. GANs), trained sequentially.

---

**Input:** Training sample $S_N := \{X_1, \ldots, X_N\}$.
**Output:** Mixture generative model $G = G_T$.
    Train vanilla GAN $G_1 = \mathrm{GAN}(S_N, W_1)$ with a uniform weight $W_1 = (1/N, \ldots, 1/N)$ over the training points
    **for** $t = 2, \ldots, T$ **do**
        *#Choose the overall weight of the next mixture component*
        $\beta_t = \mathrm{ChooseMixtureWeight}(t)$
        *#Update the weight of each training example*
        $W_t = \mathrm{UpdateTrainingWeights}(G_{t-1}, S_N, \beta_t)$
        *#Train $t$-th "weak" component generator $G_t^c$*
        $G_t^c = \mathrm{GAN}(S_N, W_t)$
        *#Update the overall generative model:*
        *#Form a mixture of $G_{t-1}$ and $G_t^c$.*
        $G_t = (1 - \beta_t)G_{t-1} + \beta_t G_t^c$
    **end for**

---

to tackle the missing mode problem. Thus, we do not aim at improving the original GAN or any other generative algorithm. We rather propose and analyse a meta-algorithm that can be used on top of any of them. This meta-algorithm is similar in spirit to AdaBoost in the sense that each iteration corresponds to learning a "weak" generative model (e.g., GAN) with respect to a re-weighted data distribution. The weights change over time to focus on the "hard" examples, i.e. those that the mixture has not been able to properly generate so far.

**Related Work** Several authors [6, 7, 8] have proposed to use boosting techniques in the context of density estimation by incrementally adding components in the log domain. This idea was applied to GANs in [8]. A major downside of these approaches is that the resulting mixture is a product of components and sampling from such a model is nontrivial (at least when applied to GANs where the model density is not expressed analytically) and requires techniques such as Annealed Importance Sampling [9] for the normalization.

When the log likelihood can be computed, [10] proposed to use an additive mixture model. They derived the update rule via computing the steepest descent direction when adding a component with infinitesimal weight. However, their results do not apply once the weight $\beta$ becomes non-infinitesimal. In contrast, for any fixed weight of the new component our approach gives the overall optimal update (rather than just the best direction) for a specified $f$-divergence. In both theories, improvements of the mixture are guaranteed only if the new "weak" learner is still good enough (see Conditions 10&11)

Similarly, [11] studied the construction of mixtures minimizing the Kullback divergence and proposed a greedy procedure for doing so. They also proved that under certain conditions, finite mixtures can approximate arbitrary mixtures at a rate $1/k$ where $k$ is the number of components in the mixture when the weight of each newly added component is $1/k$. These results are specific to the Kullback divergence but are consistent with our more general results.

An additive procedure similar to ours was proposed in [12] but with a different re-weighting scheme, which is not motivated by a theoretical analysis of optimality conditions. On every new iteration the authors run GAN on the $k$ training examples with maximal values of the discriminator from the last iteration.

Finally, many papers investigate completely different approaches for addressing the same issue by directly modifying the training objective of an individual GAN. For instance, [5] add an autoencoding cost to the training objective of GAN, while [4] allow the generator to "look few steps ahead" when making a gradient step.

The paper is organized as follows. In Section 2 we present our main theoretical results regarding iterative optimization of mixture models under general $f$-divergences. In Section 2.4 we show that if optimization at each step is perfect, the process converges to the true data distribution at exponential rate (or even in a *finite number of steps*, for which we provide a necessary and sufficient condition). Then we show in Section 2.5 that imperfect solutions still lead to the exponential rate of convergence under certain "weak learnability" conditions. These results naturally lead to a new boosting-style iterative procedure for constructing generative models. When used with GANs, it results in our *AdaGAN* algorithm, detailed in Section 3 . Finally, we report initial empirical results in Section 4, where we compare AdaGAN with several benchmarks, including original GAN and uniform mixture of multiple independently trained GANs. Part of new theoretical results are reported without proofs, which can be found in appendices.

## 2 Minimizing $f$-divergence with Mixtures

### 2.1 Preliminaries and notations

**Generative Density Estimation** In density estimation, one tries to approximate a real data distribution $P_d$, defined over the data space $\mathcal{X}$, by a model distribution $P_{model}$. In the generative approach one builds a function $G : \mathcal{Z} \to \mathcal{X}$ that transforms a fixed probability distribution $P_Z$ (often called the *noise* distribution) over a latent space $\mathcal{Z}$ into a distribution over $\mathcal{X}$. Hence $P_{model}$ is the pushforward of $P_Z$, i.e. $P_{model}(A) = P_Z(G^{-1}(A))$. With this approach it is in general impossible to compute the density $dP_{model}(x)$ and the log-likelihood of the training data under the model, but one can easily sample from $P_{model}$ by sampling from $P_Z$ and applying $G$. Thus, to construct $G$, instead of comparing $P_{model}$ directly with $P_d$, one compares their samples. To do so, one uses a similarity measure $D(P_{model}\|P_d)$ which can be estimated from samples of those distributions, and thus approximately minimized over a class $\mathcal{G}$ of functions.

$f$**-Divergences** In order to measure the agreement between the model distribution and the true distribution we will use an $f$-divergence defined in the following way:

$$D_f(Q\|P) := \int f\left(\frac{dQ}{dP}(x)\right) dP(x) \tag{1}$$

for any pair of distributions $P, Q$ with densities $dP, dQ$ with respect to some dominating reference measure $\mu$ (we refer to Appendix D for more details about such divergences and their domain of definition). Here we assume that $f$ is convex, defined on $(0, \infty)$, and satisfies $f(1) = 0$. We will denote by $\mathcal{F}$ the set of such functions. [2]

As demonstrated in [16, 17], several commonly used symmetric $f$-divergences are *Hilbertian metrics*, which in particular means that their square root satisfies the triangle inequality. This is true for the Jensen-Shannon divergence[3], the Hellinger distance and the Total Variation among others. We will denote by $\mathcal{F}_H$ the set of functions $f$ such that $D_f$ is a Hilbertian metric.

**GAN and $f$-divergences** The original GAN algorithm [2] optimizes the following criterion:

$$\min_G \max_D \mathbb{E}_{P_d}\left[\log D(X)\right] + \mathbb{E}_{P_Z}\left[\log(1 - D(G(Z)))\right], \tag{2}$$

where $D$ and $G$ are two functions represented by neural networks. This optimization is performed on a pair of samples (a training sample from $P_d$ and a "fake" sample from $P_Z$), which corresponds to approximating the above criterion by using the empirical distributions. In the non-parametric limit for $D$, this is equivalent to minimizing the Jensen-Shannon divergence [2]. This point of view can be generalized to any other $f$-divergence [13]. Because of this strong connection between adversarial

training of generative models and minimization of $f$-divergences, we cast the results of this section into the context of general $f$-divergences.

**Generative Mixture Models** In order to model complex data distributions, it can be convenient to use a mixture model of the following form: $P_{model}^T := \sum_{i=1}^{T} \alpha_i P_i$, where $\alpha_i \geq 0$, $\sum_i \alpha_i = 1$, and each of the $T$ components is a generative density model. This is natural in the generative context, since sampling from a mixture corresponds to a two-step sampling, where one first picks the mixture component (according to the multinomial distribution with parameters $\alpha_i$) and then samples from it. Also, this allows to construct complex models from simpler ones.

## 2.2 Incremental Mixture Building

We restrict ourselves to the case of $f$-divergences and assume that, given an i.i.d. sample from any unknown distribution $P$, we can construct a simple model $Q \in \mathcal{G}$ which approximately minimizes[4]

$$\min_{Q \in \mathcal{G}} D_f(Q \,\|\, P). \tag{3}$$

Instead of modelling the data with a single distribution, we now want to model it with a mixture of distributions $P_i$, where each $P_i$ is obtained by a training procedure of the form (3) with (possibly) different target distributions $P$ for each $i$. A natural way to build a mixture is to do it incrementally: we train the first model $P_1$ to minimize $D_f(P_1 \,\|\, P_d)$ and set the corresponding weight to $\alpha_1 = 1$, leading to $P_{model}^1 = P_1$. Then after having trained $t$ components $P_1, \ldots, P_t \in \mathcal{G}$ we can form the $(t+1)$-st mixture model by adding a new component $Q$ with weight $\beta$ as follows:

$$P_{model}^{t+1} := \sum_{i=1}^{t} (1-\beta)\alpha_i P_i + \beta Q. \tag{4}$$

where $\beta \in [0,1]$ and $Q \in \mathcal{G}$ is computed by minimizing:

$$\min_Q D_f((1-\beta)P_g + \beta Q \,\|\, P_d), \tag{5}$$

where we denoted $P_g := P_{model}^t$ the current generative mixture model before adding the new component. We do not expect to find the optimal $Q$ that minimizes (5) at each step, but we aim at constructing some $Q$ that slightly improves our current approximation of $P_d$, i.e. such that for $c < 1$

$$D_f((1-\beta)P_g + \beta Q \,\|\, P_d) \leq c \cdot D_f(P_g \,\|\, P_d). \tag{6}$$

This greedy approach has a significant drawback in practice. As we build up the mixture, we need to make $\beta$ decrease (as $P_{model}^t$ approximates $P_d$ better and better, one should make the correction at each step smaller and smaller). Since we are approximating (5) using samples from both distributions, this means that the sample from the mixture will only contain a fraction $\beta$ of examples from $Q$. So, as $t$ increases, getting meaningful information from a sample so as to tune $Q$ becomes harder and harder (the information is "diluted"). To address this issue, we propose to optimize an upper bound on (5) which involves a term of the form $D_f(Q \,\|\, R)$ for some distribution $R$, which can be computed as a re-weighting of the original data distribution $P_d$. This procedure is reminiscent of the AdaBoost algorithm [18], which combines multiple *weak* predictors into one *strong* composition. On each step AdaBoost adds new predictor to the current composition, which is trained to minimize the binary loss on the re-weighted training set. The weights are constantly updated to bias the next weak learner towards "hard" examples, which were incorrectly classified during previous stages.

In the following we will analyze the properties of (5) and derive upper bounds that provide practical optimization criteria for building the mixture. We will also show that under certain assumptions, the minimization of the upper bound leads to the optimum of the original criterion.

## 2.3 Upper Bounds

We provide two upper bounds on the divergence of the mixture in terms of the divergence of the additive component $Q$ with respect to some reference distribution $R$.

**Lemma 1** *Given two distributions $P_d, P_g$ and some $\beta \in [0,1]$, then, for any $Q$ and $R$, and $f \in \mathcal{F}_H$:*

$$\sqrt{D_f((1-\beta)P_g + \beta Q \,\|\, P_d)} \leq \sqrt{\beta D_f(Q \,\|\, R)} + \sqrt{D_f((1-\beta)P_g + \beta R \,\|\, P_d)}. \qquad (7)$$

*If, more generally, $f \in \mathcal{F}$, but $\beta dR \leq dP_d$, then:*

$$D_f((1-\beta)P_g + \beta Q \,\|\, P_d) \leq \beta D_f(Q \,\|\, R) + (1-\beta)D_f\left(P_g \,\bigg\|\, \frac{P_d - \beta R}{1-\beta}\right). \qquad (8)$$

We can thus exploit those bounds by introducing some well-chosen distribution $R$ and then minimizing them with respect to $Q$. A natural choice for $R$ is a distribution that minimizes the last term of the upper bound (which does not depend on $Q$). Our main result indicates the shape of the distributions minimizing the right-most terms in those bounds.

**Theorem 1** *For any $f$-divergence $D_f$, with $f \in \mathcal{F}$ and $f$ differentiable, any fixed distributions $P_d, P_g$, and any $\beta \in (0,1]$, the minimizer of (5) over all probability distributions $\mathbb{P}$ has density*

$$dQ_\beta^*(x) = \frac{1}{\beta}\left(\lambda^* dP_d(x) - (1-\beta)dP_g(x)\right)_+ = \frac{dP_d}{\beta}\left(\lambda^* - (1-\beta)\frac{dP_g}{dP_d}\right)_+. \qquad (9)$$

*for the unique $\lambda^* \in [\beta, 1]$ satisfying $\int dQ_\beta^* = 1$. Also, $\lambda^* = 1$ if and only if $P_d((1-\beta)dP_g > dP_d) = 0$, which is equivalent to $\beta dQ_\beta^* = dP_d - (1-\beta)dP_g$.*

**Theorem 2** *Given two distributions $P_d, P_g$ and some $\beta \in (0,1]$, assume $P_d(dP_g = 0) < \beta$. Let $f \in \mathcal{F}$. The problem*

$$\min_{Q : \beta dQ \leq dP_d} D_f\left(P_g \,\bigg\|\, \frac{P_d - \beta Q}{1-\beta}\right)$$

*has a solution with the density $dQ_\beta^\dagger(x) = \frac{1}{\beta}\left(dP_d(x) - \lambda^\dagger(1-\beta)dP_g(x)\right)_+$ for the unique $\lambda^\dagger \geq 1$ that satisfies $\int dQ_\beta^\dagger = 1$.*

Surprisingly, in both Theorems 1 and 2, the solutions do not depend on the choice of the function $f$, which means that the solution is the same for *any* $f$-divergence[5]. Note that $\lambda^*$ is implicitly defined by a fixed-point equation. In Section 3 we will show how it can be computed efficiently in the case of empirical distributions.

### 2.4 Convergence Analysis for Optimal Updates

In previous section we derived analytical expressions for the distributions $R$ minimizing last terms in upper bounds (8) and (7). Assuming $Q$ can perfectly match $R$, i.e. $D_f(Q \,\|\, R) = 0$, we are now interested in the convergence of the mixture (4) to the true data distribution $P_d$ when $Q = Q_\beta^*$ or $Q = Q_\beta^\dagger$. We start with simple results showing that adding $Q_\beta^*$ or $Q_\beta^\dagger$ to the current mixture would yield a strict improvement of the divergence.

**Lemma 2 (Property 6: exponential improvements)** *Under the conditions of Theorem 1, we have*

$$D_f\big((1-\beta)P_g + \beta Q_\beta^* \,\big\|\, P_d\big) \leq D_f\big((1-\beta)P_g + \beta P_d \,\big\|\, P_d\big) \leq (1-\beta)D_f(P_g \,\|\, P_d).$$

*Under the conditions of Theorem 2, we have*

$$D_f\left(P_g \,\bigg\|\, \frac{P_d - \beta Q_\beta^\dagger}{1-\beta}\right) \leq D_f(P_g \,\|\, P_d) \text{ and } D_f\big((1-\beta)P_g + \beta Q_\beta^\dagger \,\big\|\, P_d\big) \leq (1-\beta)D_f(P_g \,\|\, P_d).$$

Imagine repeatedly adding $T$ new components to the current mixture $P_g$, where on every step we use the same weight $\beta$ and choose the components described in Theorem 1. In this case Lemma 2 guarantees that the original objective value $D_f(P_g \,\|\, P_d)$ would be reduced at least to $(1-\beta)^T D_f(P_g \,\|\, P_d)$.

This exponential rate of convergence, which at first may look surprisingly good, is simply explained by the fact that $Q_\beta^*$ depends on the true distribution $P_d$, which is of course unknown.

Lemma 2 also suggests setting $\beta$ as large as possible since we assume we can compute the optimal mixture component (which for $\beta = 1$ is $P_d$). However, in practice we may prefer to keep $\beta$ relatively small, preserving what we learned so far through $P_g$: for instance, when $P_g$ already covered part of the modes of $P_d$ and we want $Q$ to cover the remaining ones. We provide further discussions on choosing $\beta$ in Section 3.

## 2.5 Weak to Strong Learnability

In practice the component $Q$ that we add to the mixture is not exactly $Q_\beta^*$ or $Q_\beta^\dagger$, but rather an approximation to them. In this section we show that if this approximation is good enough, then we retain the property (6) (exponential improvements).

Looking again at Lemma 1 we notice that the first upper bound is less tight than the second one. Indeed, take the optimal distributions provided by Theorems 1 and 2 and plug them back as $R$ into the upper bounds of Lemma 1. Also assume that $Q$ can match $R$ exactly, i.e. $D_f(Q \,\|\, R) = 0$. In this case both sides of (7) are equal to $D_f((1 - \beta)P_g + \beta Q_\beta^* \,\|\, P_d)$, which is the optimal value for the original objective (5). On the other hand, (8) does not become an equality and the r.h.s. is not the optimal one for (5). However, earlier we agreed that our aim is to reach the modest goal (6) and next we show that this is indeed possible. Corollaries 1 and 2 provide sufficient conditions for strict improvements when we use the upper bounds (8) and (7) respectively.

**Corollary 1** *Given $P_d$, $P_g$, and some $\beta \in (0, 1]$, assume $P_d\left(\frac{dP_g}{dP_d} = 0\right) < \beta$. Let $Q_\beta^\dagger$ be as defined in Theorem 2. If $Q$ is such that*

$$D_f(Q \,\|\, Q_\beta^\dagger) \leq \gamma D_f(P_g \,\|\, P_d) \tag{10}$$

*for $\gamma \in [0, 1]$, then $D_f((1 - \beta)P_g + \beta Q \,\|\, P_d) \leq (1 - \beta(1 - \gamma))D_f(P_g \,\|\, P_d)$.*

**Corollary 2** *Let $f \in \mathcal{F}_H$. Take any $\beta \in (0, 1]$, $P_d$, $P_g$, and let $Q_\beta^*$ be as defined in Theorem 1. If $Q$ is such that*

$$D_f(Q \,\|\, Q_\beta^*) \leq \gamma D_f(P_g \,\|\, P_d) \tag{11}$$

*for some $\gamma \in [0, 1]$, then $D_f((1 - \beta)P_g + \beta Q \,\|\, P_d) \leq C_{\gamma,\beta} \cdot D_f(P_g \,\|\, P_d)$, where $C_{\gamma,\beta} = \left(\sqrt{\gamma\beta} + \sqrt{1 - \beta}\right)^2$ is strictly smaller than 1 as soon as $\gamma < \beta/4$ (and $\beta > 0$).*

Conditions 10 and 11 may be compared to the "weak learnability" condition of AdaBoost. As long as our weak learner is able to solve the surrogate problem (3) of matching respectively $Q_\beta^\dagger$ or $Q_\beta^*$ accurately enough, the original objective (5) is guaranteed to decrease as well. It should be however noted that Condition 11 with $\gamma < \beta/4$ is perhaps too strong to call it "weak learnability". Indeed, as already mentioned before, the weight $\beta$ is expected to decrease to zero as the number of components in the mixture distribution $P_g$ increases. This leads to $\gamma \to 0$, making it harder to meet Condition 11. This obstacle may be partially resolved by the fact that we will use a GAN to fit $Q$, which corresponds to a relatively rich[6] class of models $\mathcal{G}$ in (3). In other words, our weak learner is not so weak. On the other hand, Condition 10 of Corollary 1 is milder. No matter what $\gamma \in [0, 1]$ and $\beta \in (0, 1]$ are, the new component $Q$ is guaranteed to strictly improve the objective functional. This comes at the price of the additional condition $P_d(dP_g/dP_d = 0) < \beta$, which asserts that $\beta$ should be larger than the mass of true data $P_d$ missed by the current model $P_g$. We argue that this is a rather reasonable condition: if $P_g$ misses many modes of $P_d$ we would prefer assigning a relatively large weight $\beta$ to the new component $Q$. However, in practice, both Conditions 10 and 11 are difficult to check. A rigorous analysis of situations when they are guaranteed is a direction for future research.

# 3 AdaGAN

We now describe the functions *ChooseMixtureWeight* and *UpdateTrainingWeights* of Algorithm 1. The complete AdaGAN meta-algorithm with the details of *UpdateTrainingWeight* and *ChooseMixtureWeight*, is summarized in Algorithm 3 of Appendix A.

**UpdateTrainingWeights** At each iteration we add a new component $Q$ to the current mixture $P_g$ with weight $\beta$. The component $Q$ should approach the "optimal target" $Q_\beta^*$ provided by (9) in Theorem 1. This distribution depends on the density ratio $dP_g/dP_d$, which is not directly accessible, but it can be estimated using adversarial training. Indeed, we can train a separate *mixture discriminator* $D_M$ to distinguish between samples from $P_d$ and samples from the current mixture $P_g$. It is known [13] that for an arbitrary $f$-divergence, there exists a corresponding function $h$ such that the values of the optimal discriminator $D_M$ are related to the density ratio by

$$\frac{dP_g}{dP_d}(x) = h\big(D_M(x)\big). \tag{12}$$

We can replace $dP_g(x)/dP_d(x)$ in (9) with $h\big(D_M(x)\big)$. For the Jensen-Shannon divergence, used by the original GAN algorithm, $h(z) = \frac{1-z}{z}$. In practice, when we compute $dQ_\beta^*$ on the training sample $S_N = (X_1, \dots, X_N)$, each example $X_i$ receives weight

$$w_i = \frac{1}{\beta N}\big(\lambda^* - (1-\beta)h(d_i)\big)_+, \quad \text{where} \quad d_i = D_M(X_i). \tag{13}$$

The only remaining task is to determine $\lambda^*$. As the weights $w_i$ in (13) must sum to 1, we get:

$$\lambda^* = \frac{\beta}{\sum_{i \in \mathcal{I}(\lambda^*)} p_i} \left(1 + \frac{(1-\beta)}{\beta} \sum_{i \in \mathcal{I}(\lambda^*)} p_i h(d_i)\right) \tag{14}$$

where $\mathcal{I}(\lambda) := \{i : \lambda > (1-\beta)h(d_i)\}$. To find $\mathcal{I}(\lambda^*)$, we sort $h(d_i)$ in increasing order: $h(d_1) \leq \dots \leq h(d_N)$. Then $\mathcal{I}(\lambda^*)$ is a set consisting of the first $k$ indices. We then successively test all $k$-s until the $\lambda$ given by (14) verifies $(1-\beta)h(d_k) < \lambda \leq (1-\beta)h(d_{k+1})$. This procedure is guaranteed to converge by Theorem 1. It is summarized in Algorithm 2 of Appendix A

**ChooseMixtureWeight** For every $\beta$ there is an optimal re-weighting scheme with weights given by (13). If the GAN could perfectly approximate its target $Q_\beta^*$, then choosing $\beta = 1$ would be optimal, because $Q_1^* = P_d$. But in practice, GANs cannot do that. So we propose to choose $\beta$ heuristically by imposing that each generator of the final mixture model has same weight. This yields $\beta_t = 1/t$, where $t$ is the iteration index. Other heuristics are proposed in Appendix B, but did not lead to any significant difference.

**The optimal discriminator** In practice it is of course hard to find the optimal discriminator $D_M$ achieving the global maximum of the variational representation for the f-divergence and verifying (12). For the JS-divergence this would mean that $D_M$ is the classifier achieving minimal expected cross-entropy loss in the binary classification between $P_g$ and $P_d$. In practice, we observed that the reweighting (13) leads to the desired property of emphasizing at least some of the missing modes as long as $D_M$ distinguishes reasonably between data points already covered by the current model $P_g$ and those which are still missing. We found an early stopping (while training $D_M$) sufficient to achieve this. In the *worst case*, when $D_M$ overfits and returns 1 for all true data points, the reweighting simply leads to the uniform distribution over the training set.

# 4 Experiments

We ran AdaGAN[7] on toy datasets, for which we can interpret the missing modes in a clear and reproducible way, and on MNIST, which is a high-dimensional dataset. The goal of these experiments *was not* to evaluate the visual quality of individual sample points, but to demonstrate that the re-weighting scheme of AdaGAN promotes diversity and effectively covers the missing modes.

**Toy Datasets**   Our target distribution is a mixture of isotropic Gaussians over $\mathcal{R}^2$. The distances between the means are large enough to roughly avoid overlaps between different Gaussian components. We vary the number of modes to test how well each algorithm performs when there are fewer or more expected modes. We compare the baseline GAN algorithm with AdaGAN variations, and with other meta-algorithms that all use the same underlying GAN procedure. For details on these algorithms and on the architectures of the underlying generator and discriminator, see Appendix B.

To evaluate how well the generated distribution matches the target distribution, we use a *coverage* metric $C$. We compute the probability mass of the true data "covered" by the model $P_{model}$. More precisely, we compute $C := P_d(dP_{model} > t)$ with $t$ such that $P_{model}(dP_{model} > t) = 0.95$. This metric is more interpretable than the likelihood, making it easier to assess the difference in performance of the algorithms. To approximate the density of $P_{model}$ we use a kernel density estimation, where the bandwidth is chosen by cross validation. We repeat the run 35 times with the same parameters (but different random seeds). For each run, the learning rate is optimized using a grid search on a validation set. We report the median over those multiple runs, and the interval corresponding to the 5% and 95% percentiles.

Figure 2 summarizes the performance of algorithms as a function of the number of iterations $T$. Both the ensemble and the boosting approaches significantly outperform the vanilla GAN and the "best of $T$" algorithm. Interestingly, the improvements are significant even after just one or two additional iterations ($T = 2$ or 3). Our boosting approach converges much faster. In addition, its variance is much lower, improving the likelihood that a given run gives good results. On this setup, the vanilla GAN approach has a significant number of catastrophic failures (visible in the lower bounds of the intervals). Further empirical results are available in Appendix B, where we compared AdaGAN variations to several other baseline meta-algorithms in more details (Table 1) and combined AdaGAN with the unrolled GANs (UGAN) [4] (Figure 3). Interestingly, Figure 3 shows that AdaGAN ran with UGAN outperforms the vanilla UGAN on the toy datasets, demonstrating the advantage of using AdaGAN as a way to further improve the mode coverage of any existing GAN implementations.

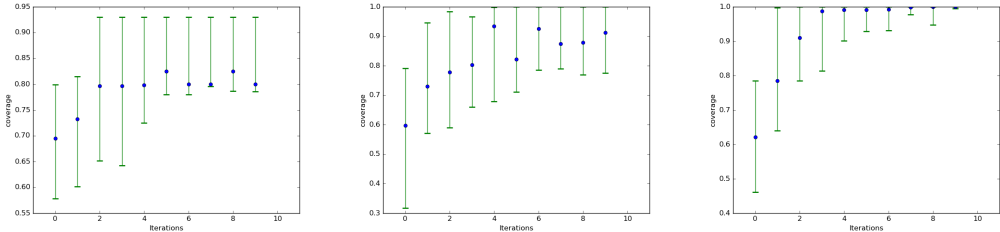

Figure 1: Coverage $C$ of the true data by the model distribution $P_{model}^T$, as a function of iterations $T$. Experiments correspond to the data distribution with 5 modes. Each blue point is the median over 35 runs. Green intervals are defined by the 5% and 95% percentiles (see Section 4). Iteration 0 is equivalent to one vanilla GAN. The left plot corresponds to taking the best generator out of $T$ runs. The middle plot is an "ensemble" GAN, simply taking a uniform mixture of $T$ independently trained GAN generators. The right plot corresponds to our boosting approach (AdaGAN), with $\beta_t = 1/t$.

**MNIST and MNIST3**   We ran experiments both on the original MNIST and on the 3-digit MNIST (MNIST3) [5, 4] dataset, obtained by concatenating 3 randomly chosen MNIST images to form a 3-digit number between 0 and 999. According to [5, 4], MNIST contains 10 modes, while MNIST3 contains 1000 modes, and these modes can be detected using the pre-trained MNIST classifier. We combined AdaGAN both with simple MLP GANs and DCGANs [19]. We used $T \in \{5, 10\}$, tried models of various sizes and performed a reasonable amount of hyperparameter search.

Similarly to [4, Sec 3.3.1] we failed to reproduce the missing modes problem for MNIST3 reported in [5] and found that simple GAN architectures are capable of generating all 1000 numbers. The authors of [4] proposed to artificially introduce the missing modes again by limiting the generators' flexibility. In our experiments, GANs trained with the architectures reported in [4] were often generating poorly looking digits. As a result, the pre-trained MNIST classifier was outputting random labels, which again led to full coverage of the 1000 numbers. We tried to threshold the confidence of the pre-trained classifier, but decided that this metric was too ad-hoc.

For MNIST we noticed that the re-weighted distribution was often concentrating its mass on digits having very specific strokes: on different rounds it could highlight thick, thin, vertical, or diagonal digits, indicating that these traits were underrepresented in the generated samples (see Figure 2). This suggests that AdaGAN does a reasonable job at picking up different modes of the dataset, but also that there are more than 10 modes in MNIST (and more than 1000 in MNIST3). It is not clear how to evaluate the quality of generative models in this context.

We also tried to use the "inversion" metric discussed in Section 3.4.1 of [4]. For MNIST3 we noticed that a single GAN was capable of reconstructing most of the training points *very* accurately both visually and in the $\ell_2$-reconstruction sense. The "inversion" metric tests whether the trained model can generate certain examples or not, but unfortunately it does not take into account *the probabilities* of doing so.

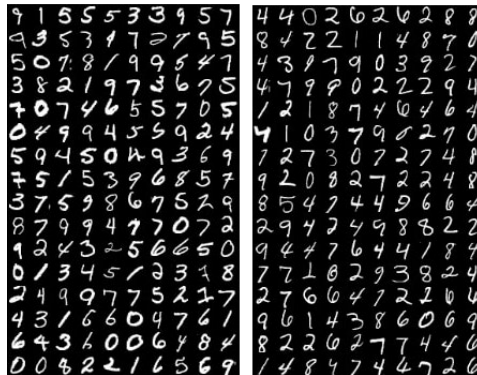

Figure 2: Digits from the MNIST dataset corresponding to the smallest (**left**) and largest (**right**) weights, obtained by the AdaGAN procedure (see Section 3) in one of the runs. Bold digits (left) are already covered and next GAN will concentrate on thin (right) digits.

## 5   Conclusion

We studied the problem of minimizing general $f$-divergences with additive mixtures of distributions. The main contribution of this work is a detailed theoretical analysis, which naturally leads to an iterative greedy procedure. On every iteration the mixture is updated with a new component, which minimizes $f$-divergence with a re-weighted target distribution. We provided conditions under which this procedure is guaranteed to converge to the target distribution at an exponential rate. While our results can be combined with any generative modelling techniques, we focused on GANs and provided a boosting-style algorithm *AdaGAN*. Preliminary experiments show that AdaGAN successfully produces a mixture which iteratively covers the missing modes.

## Footnotes

[1]Note that the term "mixture" should not be interpreted to imply that each component models only one mode: the models to be combined into a mixture can themselves cover multiple modes.

[2]Examples of $f$-divergences include the Kullback-Leibler divergence (obtained for $f(x) = x \log x$) and Jensen-Shannon divergence ($f(x) = -(x+1) \log \frac{x+1}{2} + x \log x$). Other examples can be found in [13]. For further details we refer to Section 1.3 of [14] and [15].

[3]which means such a property can be used in the context of the original GAN algorithm.

[4]One example of such a setting is running GANs.

[5]in particular, by replacing $f$ with $f^\circ(x) := xf(1/x)$, we get the same solution for the criterion written in the other direction. Hence the order in which we write the divergence does not matter and the optimal solution is optimal for both orders.

[6]The hardness of meeting Condition 11 of course largely depends on the class of models $\mathcal{G}$ used to fit $Q$ in (3). For now we ignore this question and leave it for future research.

[7]Code available online at `https://github.com/tolstikhin/adagan`

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
