[Supplementary Material · adagan2017nips_supp.pdf]

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

# A   Algorithms

---
**ALGORITHM 2** Determining $\lambda^*$

---
Sort the values $h(d_i)$ in increasing order

Initialize $\lambda \leftarrow \frac{\beta}{p_1}\left(1 + \frac{1-\beta}{\beta}p_1 h(d_1)\right)$ and $k \leftarrow 1$

**while** $(1-\beta)h(d_k) \geq \lambda$ **do**

   $k \leftarrow k + 1$

   $\lambda \leftarrow \frac{\beta}{\sum_{i=1}^{k} p_i}\left(1 + \frac{(1-\beta)}{\beta}\sum_{i=1}^{k} p_i h(d_i)\right)$

**end while**

---

---
**ALGORITHM 3**

AdaGAN, a meta-algorithm to construct a "strong" mixture of $T$ individual GANs, trained sequentially. The mixture weight schedule ChooseMixtureWeight should be provided by the user (see 3). This is an instance of the high level Algorithm 1, instantiating UpdateTrainingWeights.

---
**Input:** Training sample $S_N := \{X_1, \ldots, X_N\}$.
**Output:** Mixture generative model $G = G_T$.

  Train vanilla GAN: $G_1 = \mathrm{GAN}(S_N)$

  **for** $t = 2, \ldots, T$ **do**

    *#Choose a mixture weight for the next component*

    $\beta_t = \mathrm{ChooseMixtureWeight}(t)$

    *#Compute the new weights of the training examples (UpdateTrainingWeights)*

    *#Compute the discriminator between the original (unweighted) data and the current mixture*

    $G_{t-1}$

    $D \leftarrow DGAN(S_N, G_{t-1})$

    *#Compute $\lambda^*$ using Algorithm 2*

    $\lambda^* \leftarrow \lambda(\beta_t, D)$

    *#Compute the new weight for each example*

    **for** $i = 1, \ldots, N$ **do**

      $W_t^i = \frac{1}{N\beta_t}\left(\lambda^* - (1-\beta_t)h(D(X_i))\right)_+$

    **end for**

    *#Train $t$-th "weak" component generator $G_t^c$*

    $G_t^c = \mathrm{GAN}(S_N, W_t)$

    *#Update the overall generative model*

    *#Notation below means forming a mixture of $G_{t-1}$ and $G_t^c$.*

    $G_t = (1-\beta_t)G_{t-1} + \beta_t G_t^c$

  **end for**

---

# B   Details on the Toy Experiments

**GAN architectures** In all our experiments, the GAN's generator uses the latent space $\mathcal{Z} = \mathcal{R}^5$, and two ReLU hidden layers, of size 10 and 5 respectively. The corresponding discriminator has two ReLU hidden layers of size 20 and 10 respectively. We use 64k training examples, and 15 epochs, which is enough compared to the small scale of the problem. The optimizer is a simple SGD: Adam was also tried but gave slightly less stable results. All networks converge properly and overfitting is never an issue.

**Details on the tested algorithms and more tests** In our experiments, we compared the following algorithms:

   – The baseline GAN algorithm, called **Vanilla GAN** in the results.

|  | $Modes : 1$ | $Modes : 2$ | $Modes : 3$ | $Modes : 5$ | $Modes : 10$ |
|---|---|---|---|---|---|
| Vanilla | 0.97 (0.9; 1.0) | 0.88 (0.4; 1.0) | 0.63 (0.5; 1.0) | 0.72 (0.5; 0.8) | 0.59 (0.2; 0.7) |
| Best of T (T=3) | 0.99 (1.0; 1.0) | 0.96 (0.9; 1.0) | 0.91 (0.7; 1.0) | 0.80 (0.7; 0.9) | 0.70 (0.6; 0.8) |
| Best of T (T=10) | 0.99 (1.0; 1.0) | 0.99 (1.0; 1.0) | 0.98 (0.8; 1.0) | 0.80 (0.8; 0.9) | 0.71 (0.7; 0.8) |
| Ensemble (T=3) | 0.99 (1.0; 1.0) | 0.98 (0.9; 1.0) | 0.93 (0.8; 1.0) | 0.78 (0.6; 1.0) | 0.80 (0.6; 1.0) |
| Ensemble (T=10) | 1.00 (1.0; 1.0) | 0.99 (1.0; 1.0) | 1.00 (1.0; 1.0) | 0.91 (0.8; 1.0) | 0.89 (0.7; 1.0) |
| TopKLast0.5 (T=3) | 0.98 (0.9; 1.0) | 0.98 (0.9; 1.0) | 0.95 (0.9; 1.0) | 0.95 (0.8; 1.0) | 0.86 (0.6; 0.9) |
| TopKLast0.5 (T=10) | 0.99 (1.0; 1.0) | 0.98 (0.9; 1.0) | 0.98 (1.0; 1.0) | 0.99 (0.8; 1.0) | 1.00 (0.8; 1.0) |
| Boosted (T=3) | 0.99 (1.0; 1.0) | 0.99 (0.9; 1.0) | 0.98 (0.9; 1.0) | 0.91 (0.8; 1.0) | 0.86 (0.7; 1.0) |
| Boosted (T=10) | 1.00 (1.0; 1.0) | 1.00 (1.0; 1.0) | 1.00 (1.0; 1.0) | 1.00 (1.0; 1.0) | 1.00 (1.0; 1.0) |

Table 1: Performance of the different algorithms on varying number of mixtures of Gaussians. The reported score is the coverage $C$, probability mass of $P_d$ covered by the $5th$ percentile of $P_g$ defined in Section 4. The reported scores are the median and interval defined by the 5% and 95% percentile (in parenthesis) (see Section 4), over 35 runs for each setting. Both the ensemble and the boosting approaches significantly outperform the vanilla GAN even with just three iterations (i.e. just two additional components). The boosting approach converges faster to the optimal coverage and with smaller variance.

(a) The best model out of $T$ runs of GAN, that is: run $T$ GAN instances independently, then take the run that performs best on a validation set. This gives an additional baseline with similar computational complexity as the ensemble approaches. Note that the selection of the best run is done on the reported target metric (see below), rather than on the internal metric. As a result this baseline is slightly overestimated. This procedure is called **Best of T** in the results.

(b) A mixture of $T$ GAN generators, trained independently, and combined with equal weights (the "bagging" approach). This procedure is called **Ensemble** in the results.

– A mixture of GAN generators, trained sequentially with different choices of data re-weighting:

(c) The AdaGAN algorithm (Algorithm 1), with $\beta = 1/t$. Thus each component will have the same weight in the resulting mixture (see Equation 3). This procedure is called **Boosted** in the results.

– The AdaGAN algorithm (Algorithm 1), but with a constant $\beta$, exploring several values. This procedure is called for example **Beta0.3** for $\beta = 0.3$ in the results. Note that in this setting, not all components of the mixture have the same weight.

– Reweighting similar to "Cascade GAN" from [12], i.e. keeping the top $r$ fraction of examples, based on the discriminator corresponding to the *previous* generator. This procedure is called for example **TopKLast0.3** for $r = 0.3$.

– Keep the top $r$ fraction of examples, based on the discriminator corresponding to *the mixture of all previous* generators. This procedure is called for example **TopK0.3** for $r = 0.3$.

The left, middle, and right panels in Figure 2 of Section 4 respectively correspond to the settings (a), (b) and (c).

**Experiments with unrolled GAN** To illustrate the 'meta-algorithm aspect' of AdaGAN, we also performed experiments with an unrolled GAN (UGAN) [4] instead of a GAN as the base generator. We trained the GANs both with the Jensen-Shannon objective (2), and with its modified version proposed in [2] (and often considered as the baseline GAN), where $\log(1 - D(G(Z)))$ is replaced by $-\log(D(G(Z)))$. We use the same network architecture as in the other toy experiments. Figure 3 illustrates our results. We find that AdaGAN works with all UGAN algorithms. Note that, where the usual GAN updates the generator and the discriminator once, an UGAN with 5 unrolling steps updates the generator once and the discriminator $1 + 5$, i.e. 6 times (and then rolls back 5 steps). Thus, in terms of computation time, training 1 single UGAN roughly corresponds to doing 3 steps of AdaGAN with a usual GAN. In that sense, Figure 3 shows that AdaGAN (with a usual GAN)

Figure 3: Comparison of AdaGAN ran with a GAN (top row) and with an unrolled GAN with 5 unrolling steps [4] (bottom). Coverage $C$ of the true data by the model distribution $P_{model}^T$, as a function of iterations $T$. Experiments are similar to those of Figure 2, but with 10 modes. Left figures used the Jensen-Shannon objective (2), while right figures used the modified objective originally proposed by [2]. In terms of computation time, one step of AdaGAN with unrolled GAN corresponds to roughly 3 steps of AdaGAN with a usual GAN.

significantly outperforms a single unrolled GAN ($T = 1$ on bottom pictures). Also note that AdaGAN ran with UGAN outperforms a single UGAN and keeps improving its performance as we increase the number of iterations. Additionally, we note that using the Jensen-Shannon objective (rather than the modified version) seems to have some mode-regularizing effect.

## C  Details for AdaGAN on MNIST

**GAN Architecture**  We ran AdaGAN on MNIST (28x28 pixel images) using (de)convolutional networks with batch normalizations and leaky ReLu. The latent space has dimension 100. We used the following architectures:

Generator:    100 x 1 x 1 –> fully connected –> 7 x 7 x 16 –> deconv –> 14 x 14 x 8 –>
              –> deconv –> 28 x 28 x 4 –> deconv –> 28 x 28 x 1
Discriminator:    28 x 28 x 1 –> conv –> 14 x 14 x 16 –> conv –> 7 x 7 x 32 –>
              –> fully connected –> 1

where each arrow consists of a leaky ReLu (with 0.3 leak) followed by a batch normalization, conv and deconv are convolutions and transposed convolutions with 5x5 filters, and fully connected are linear layers with bias. The distribution over $\mathcal{Z}$ is uniform over the unit box. We use the Adam optimizer with $\beta_1 = 0.5$, with 2 G steps for 1 D step and learning rates 0.005 for G, 0.001 for D, and 0.0001 for the classifier C that does the reweighting of digits. We optimized D and G over 200 epochs and C over 5 epochs, using the original Jensen-Shannon objective (2), without the log trick, with no unrolling and with minibatches of size 128.

**Empirical observations**  Although we could not find any appropriate metric to measure the increase of diversity promoted by AdaGAN, we observed that the re-weighting scheme indeed focuses on digits with very specific strokes. In Figure 4 for example, we see that after 1 AdaGAN step, the generator produces overly thick digits (top left image). Thus AdaGAN puts small weights on the thick digits of the dataset (bottom left) and high weights on the thin ones (bottom right). After the next step, the new GAN produces both thick and thin digits.

Figure 4: AdaGAN on MNIST. Bottom row are true MNIST digits with smallest (left) and highest (right) weights after re-weighting at the end of the first AdaGAN step. Those with small weight are thick and resemble those generated by the GAN after the first AdaGAN step (top left). After training with the re-weighted dataset during the second iteration of AdaGAN, the new mixture produces more thin digits (top right).

## D    Details on $f$-divergences

The integral in (1) is well defined (but may take infinite values) even if $P(dQ = 0) > 0$ or $Q(dP = 0) > 0$. In this case the integral is understood as

$$D_f(Q\|P) = \int f(dQ/dP) \mathbf{1}_{[dP(x)>0, dQ(x)>0]} dP(x)$$
$$+ f(0)P(dQ = 0) + f^{\circ}(0)Q(dP = 0),$$

where both $f(0)$ and $f^{\circ}(0)$ may take value $\infty$ [14]. This is especially important in case of GAN, where it is impossible to constrain $P_{model}$ to be absolutely continuous with respect to $P_d$ or vice versa.

## E    Refinement of Lemma 2

If the ratio $dP_g/dP_d$ is almost surely bounded, the first inequality of Lemma 2 can be refined as follows.

**Lemma 3** *Under the conditions of Theorem 1*

$$D_f\big((1 - \beta)P_g + \beta Q_\beta^* \,\big\|\, P_d\big) \le f(\lambda^*) + \frac{f(M)(1 - \lambda^*)}{M - 1}$$

*given there exists $M > 1$ such that $P_d((1 - \beta)dP_g > MdP_d) = 0$.*

This upper bound can be tighter than that of Lemma 2 when $\lambda^*$ gets close to 1. Indeed, for $\lambda^* = 1$ the upper bound is exactly 0 and is thus tight, while the upper bound of Lemma 2 will not be zero in this case.

**Proof** We use Inequality (18) of Lemma 6 with $X = \beta$, $Y = (1 - \beta)dP_g/dP_d$, and $c = \lambda^*$. We easily verify that $X + Y = ((1 - \beta)dP_g + \beta dP_d)/dP_d$ and $\max(c, Y) = ((1 - \beta)dP_g + \beta dQ_\beta^*)/dP_d$ and both have expectation 1 with respect to $P_d$. We thus obtain:

$$D_f((1 - \beta)P_g + \beta Q_\beta^* \parallel P_d) \leq f(\lambda^*) + \frac{f(M) - f(\lambda^*)}{M - \lambda^*}(1 - \lambda^*). \tag{15}$$

Since $\lambda^* \leq 1$ and $f$ is non-increasing on $(0, 1)$ we get

$$D_f((1 - \beta)P_g + \beta Q_\beta^* \parallel P_d) \leq f(\lambda^*) + \frac{f(M)(1 - \lambda^*)}{M - 1}.$$

∎

# F    Conditions for finite steps convergence

Here we study the convergence of (5) to 0 in the case where, while performing the iterations, we use the upper bound (7) and the weight $\beta$ is fixed (i.e. the same value at each iteration). We will provide necessary and sufficient conditions for the iterative process to converge to the data distribution $P_d$ in finite number of steps. The analysis can easily be extended to a non-constant (variable) weight scheduling $\beta$. We start with the following result.

**Lemma 4** *For any $f \in \mathcal{F}$ such that $f(x) \neq 0$ for $x \neq 1$, the following conditions are equivalent:*

*(i) $P_d((1 - \beta)dP_g > dP_d) = 0$;*
*(ii) $D_f((1 - \beta)P_g + \beta Q_\beta^* \parallel P_d) = 0$.*

**Proof** The first condition is equivalent to $\lambda^* = 1$ according to Theorem 1. In this case, $(1 - \beta)P_g + \beta Q_\beta^* = P_d$, hence the divergence is 0. In the other direction, when the divergence is 0, since $f$ is strictly positive for $x \neq 1$ (keep in mind that we can always replace $f$ by $f_0$ to get a non-negative function which will be strictly positive if $f(x) \neq 0$ for $x \neq 1$), this means that with $P_d$ probability 1 we have the equality $dP_d = (1 - \beta)dP_g + \beta dQ_\beta^*$, which implies that $(1 - \beta)dP_g > dP_d$ with $P_d$ probability 1 and also $\lambda^* = 1$. ∎

This result tells that we cannot perfectly match $P_d$ by adding a new mixture component to $P_g$ as long as there are points in the space where our current model $P_g$ severely over-samples. As an example, consider an extreme case where $P_g$ puts a positive mass in a region outside of the support of $P_d$. Clearly, unless $\beta = 1$, we will not be able to match $P_d$.

We now provide the conditions for the convergence of the iterative process in a finite number of steps. The criterion is based on the ratio $dP_1/dP_d$, where $P_1$ is the first component of our mixture model.

**Corollary 3** *Take any $f \in \mathcal{F}$ such that $f(x) \neq 0$ for $x \neq 1$. Starting from $P_{model}^1 = P_1$, update the model iteratively according to $P_{model}^{t+1} = (1 - \beta)P_{model}^t + \beta Q_\beta^*$, where on every step $Q_\beta^*$ is as defined in Theorem 1 with $P_g := P_{model}^t$. In this case $D_f(P_{model}^t \parallel P_d)$ will reach 0 in a finite number of steps if and only if there exists $M > 0$ such that*

$$P_d((1 - \beta)dP_1 > MdP_d) = 0. \tag{16}$$

*When the finite convergence happens, it takes at most $-\ln\max(M, 1)/\ln(1 - \beta)$ steps.*

**Proof** From Lemma 4, it is clear that if $M \leq 1$ the convergence happens after the first update. So let us assume $M > 1$. Notice that $dP_{model}^{t+1} = (1 - \beta)dP_{model}^t + \beta dQ_\beta^* = \max(\lambda^* dP_d, (1 - \beta)dP_{model}^t)$ so that if $P_d((1 - \beta)dP_{model}^t > MdP_d) = 0$, then $P_d((1 - \beta)dP_{model}^{t+1} > M(1 - \beta)dP_d) = 0$. This proves that (16) is a sufficient condition.

Now assume the process converged in a finite number of steps. Let $P_{model}^t$ be a mixture right before the final step. Note that $P_{model}^t$ is represented by $(1-\beta)^{t-1}P_1 + (1 - (1-\beta)^{t-1})P$ for certain probability distribution $P$. According to Lemma 4 we have $P_d((1-\beta)dP_{model}^t > dP_d) = 0$. Together these two facts immediately imply (16). ∎

It is also important to keep in mind that even if (16) is not satisfied the process still converges to the true distribution at exponential rate (see Lemma 2 as well as Corollaries 1 and 2 below)

## G  Proofs

### G.1  Proof of Lemma 1

For the first inequality, we use the fact that $D_f$ is jointly convex. We write $P_d = (1-\beta)\frac{P_d - \beta R}{1-\beta} + \beta R$ which is a convex combination of two distributions when the assumptions are satisfied. The second inequality follows from using the triangle inequality for $\sqrt{D_f}$ and using convexity of $D_f$ in its first argument.

### G.2  Proof of Theorem 1

Before proving Theorem 1, we introduce two lemmas. The first one is about the determination of the constant $\lambda$, the second one is about comparing the divergences of mixtures.

**Lemma 5** *Let $P$ and $Q$ be two distributions, $\gamma \in [0,1]$ and $\lambda \in \mathcal{R}$. The function*

$$g(\lambda) := \int \left(\lambda - \gamma\frac{dQ}{dP}\right)_+ dP$$

*is nonnegative, convex, nondecreasing, satisfies $g(\lambda) \leq \lambda$, and its right derivative is given by*

$$g'_+(\lambda) = P(\lambda \cdot dP \geq \gamma \cdot dQ).$$

*The equation $g(\lambda) = 1 - \gamma$ has a solution $\lambda^*$ (unique when $\gamma < 1$) with $\lambda^* \in [1-\gamma, 1]$. Finally, if $P(dQ = 0) \geq \delta$ for a strictly positive constant $\delta$ then $\lambda^* \leq (1-\gamma)\delta^{-1}$.*

**Proof** The convexity of $g$ follows immediately from the convexity of $x \mapsto (x)_+$ and the linearity of the integral. Similarly, since $x \mapsto (x)_+$ is non-decreasing, $g$ is non-decreasing.

We define the set $\mathcal{I}(\lambda)$ as follows:

$$\mathcal{I}(\lambda) := \{x \in \mathcal{X} : \lambda \cdot dP(x) \geq \gamma \cdot dQ(x)\}.$$

Now let us consider $g(\lambda + \epsilon) - g(\lambda)$ for some small $\epsilon > 0$. This can also be written:

$$g(\lambda + \epsilon) - g(\lambda) = \int_{\mathcal{I}(\lambda)} \epsilon dP + \int_{\mathcal{I}(\lambda+\epsilon)\backslash\mathcal{I}(\lambda)} (\lambda + \epsilon)dP - \int_{\mathcal{I}(\lambda+\epsilon)\backslash\mathcal{I}(\lambda)} \gamma dQ$$

$$= \epsilon P(\mathcal{I}(\lambda)) + \int_{\mathcal{I}(\lambda+\epsilon)\backslash\mathcal{I}(\lambda)} (\lambda + \epsilon)dP - \int_{\mathcal{I}(\lambda+\epsilon)\backslash\mathcal{I}(\lambda)} \gamma dQ.$$

On the set $\mathcal{I}(\lambda + \epsilon)\backslash\mathcal{I}(\lambda)$, we have

$$(\lambda + \epsilon)dP - \gamma dQ \in [0, \epsilon].$$

So that

$$\epsilon P(\mathcal{I}(\gamma)) \leq g(\lambda + \epsilon) - g(\lambda) \leq \epsilon P(\mathcal{I}(\gamma)) + \epsilon P\big(\mathcal{I}(\lambda + \epsilon)\backslash\mathcal{I}(\lambda)\big) = \epsilon P(\mathcal{I}(\lambda + \epsilon))$$

and thus

$$\lim_{\epsilon \to 0^+} \frac{g(\lambda + \epsilon) - g(\lambda)}{\epsilon} = \lim_{\epsilon \to 0^+} P(\mathcal{I}(\lambda + \epsilon)) = P(\mathcal{I}(\lambda)).$$

This gives the expression of the right derivative of $g$. Moreover, notice that for $\lambda, \gamma > 0$

$$g'_+(\lambda) = P(\lambda \cdot dP \geq \gamma \cdot dQ) = P\left(\frac{dQ}{dP} \leq \frac{\lambda}{\gamma}\right) = 1 - P\left(\frac{dQ}{dP} > \frac{\lambda}{\gamma}\right) \geq 1 - \gamma/\lambda$$

by Markov's inequality.

It is obvious that $g(0) = 0$. By Jensen's inequality applied to the convex function $x \mapsto (x)_+$, we have $g(\lambda) \geq (\lambda - \gamma)_+$. So $g(1) \geq 1 - \gamma$. Also, $g = 0$ on $\mathcal{R}^-$ and $g \leq \lambda$. This means $g$ is continuous on $\mathcal{R}$ and thus reaches the value $1 - \gamma$ on the interval $(0, 1]$ which shows the existence of $\lambda^* \in (0, 1]$. To show that $\lambda^*$ is unique we notice that since $g(x) = 0$ on $\mathcal{R}^-$, $g$ is convex and non-decreasing, $g$ cannot be constant on an interval not containing $0$, and thus $g(x) = 1 - \gamma$ has a unique solution for $\gamma < 1$.

Also by convexity of $g$,
$$g(0) - g(\lambda^*) \geq -\lambda^* g'_+(\lambda^*),$$
which gives $\lambda^* \geq (1 - \gamma)/g'_+(\lambda^*) \geq 1 - \gamma$ since $g'_+ \leq 1$. If $P(dQ = 0) \geq \delta > 0$ then also $g'_+(0) \geq \delta > 0$. Using the fact that $g'_+$ is increasing we conclude that $\lambda^* \leq (1 - \gamma)\delta^{-1}$. ∎

Next we introduce some simple convenience lemma for comparing convex functions of random variables.

**Lemma 6** *Let $f$ be a convex function, $X, Y$ be real-valued random variables and $c \in \mathcal{R}$ be a constant such that*
$$\mathbb{E}\left[\max(c, Y)\right] = \mathbb{E}\left[X + Y\right].$$
*Then we have the following bound:*
$$\mathbb{E}\left[f(\max(c, Y))\right] \leq \mathbb{E}\left[f(X + Y)\right] - \mathbb{E}\left[X(f'(Y) - f'(c))_+\right] \leq \mathbb{E}\left[f(X + Y)\right]. \quad (17)$$
*If in addition, $Y \leq M$ a.s. for $M \geq c$, then*
$$\mathbb{E}\left[f(\max(c, Y))\right] \leq f(c) + \frac{f(M) - f(c)}{M - c}(\mathbb{E}\left[X + Y\right] - c). \quad (18)$$

**Proof** We decompose the expectation with respect to the value of the max and use the convexity of $f$:

$$
\begin{aligned}
f(X &+ Y) - f(\max(c, Y)) \\
&= \mathbb{1}_{[Y \leq c]}(f(X + Y) - f(c)) \\
&\quad + \mathbb{1}_{[Y > c]}(f(X + Y) - f(Y)) \\
&\geq \mathbb{1}_{[Y \leq c]}f'(c)(X + Y - c) + \mathbb{1}_{[Y > c]}Xf'(Y) \\
&= (1 - \mathbb{1}_{[Y > c]})Xf'(c) + f'(c)(Y - \max(c, Y)) \\
&\quad + \mathbb{1}_{[Y > c]}Xf'(Y) \\
&= f'(c)(X + Y - \max(c, Y)) \\
&\quad + \mathbb{1}_{[Y > c]}X(f'(Y) - f'(c)) \\
&= f'(c)(X + Y - \max(c, Y)) + X(f'(Y) - f'(c))_+,
\end{aligned}
$$

where we used that $f'$ is non-decreasing in the last step. Taking the expectation gives the first inequality.

For the second inequality, we use the convexity of $f$ on the interval $[c, M]$:

$$f(\max(c, Y)) \leq f(c) + \frac{f(M) - f(c)}{M - c}(\max(c, Y) - c).$$

Taking an expectation on both sides gives the second inequality. ∎

**Proof** [Theorem 1] We first apply Lemma 5 with $\gamma = 1 - \beta$ and this proves the existence of $\lambda^*$ in the interval $(\beta, 1]$, which shows that $Q^*_\beta$ is indeed well-defined as a distribution.

Then we use Inequality (17) of Lemma 6 with $X = \beta dQ/dP_d$, $Y = (1 - \beta)dP_g/dP_d$, and $c = \lambda^*$. We easily verify that $X + Y = ((1-\beta)dP_g + \beta dQ)/dP_d$ and $\max(c, Y) = ((1-\beta)dP_g + \beta dQ^*_\beta)/dP_d$ and both have expectation 1 with respect to $P_d$. We thus obtain for any distribution $Q$,
$$D_f((1 - \beta)P_g + \beta Q^*_\beta \,\|\, P_d) \leq D_f((1 - \beta)P_g + \beta Q \,\|\, P_d).$$

This proves the optimality of $Q_\beta^*$. ∎

### G.3 Proof of Theorem 2

**Lemma 7** *Let $P$ and $Q$ be two distributions, $\gamma \in (0, 1)$, and $\lambda \geq 0$. The function*

$$h(\lambda) := \int \left( \frac{1}{\gamma} - \lambda \frac{dQ}{dP} \right)_+ dP$$

*is convex, non-increasing, and its right derivative is given by $h'_+(\lambda) = -Q(1/\gamma \geq \lambda dQ(X)/dP(X))$. Denote $\Delta := P(dQ(X)/dP(X) = 0)$. Then the equation*

$$h(\lambda) = \frac{1 - \gamma}{\gamma}$$

*has no solutions if $\Delta > 1 - \gamma$, has a single solution $\lambda^\dagger \geq 1$ if $\Delta < 1 - \gamma$, and has infinitely many or no solutions when $\Delta = 1 - \gamma$.*

**Proof** The convexity of $h$ follows immediately from the convexity of $x \mapsto (a - x)_+$ and the linearity of the integral. Similarly, since $x \mapsto (a - x)_+$ is non-increasing, $h$ is non-increasing as well.

We define the set $\mathcal{J}(\lambda)$ as follows:

$$\mathcal{J}(\lambda) := \left\{ x \in \mathcal{X} : \frac{1}{\gamma} \geq \lambda \frac{dQ}{dP}(x) \right\}.$$

Now let us consider $h(\lambda) - h(\lambda + \epsilon)$ for any $\epsilon > 0$. Note that $\mathcal{J}(\lambda + \epsilon) \subseteq \mathcal{J}(\lambda)$. We can write:

$$h(\lambda) - h(\lambda + \epsilon)$$
$$= \int_{\mathcal{J}(\lambda)} \left( \frac{1}{\gamma} - \lambda \frac{dQ}{dP} \right) dP - \int_{\mathcal{J}(\lambda+\epsilon)} \left( \frac{1}{\gamma} - (\lambda + \epsilon) \frac{dQ}{dP} \right) dP$$
$$= \int_{\mathcal{J}(\lambda) \setminus \mathcal{J}(\lambda+\epsilon)} \left( \frac{1}{\gamma} - \lambda \frac{dQ}{dP} \right) dP + \int_{\mathcal{J}(\lambda+\epsilon)} \left( \epsilon \frac{dQ}{dP} \right) dP$$
$$= \int_{\mathcal{J}(\lambda) \setminus \mathcal{J}(\lambda+\epsilon)} \left( \frac{1}{\gamma} - \lambda \frac{dQ}{dP} \right) dP + \epsilon \cdot Q(\mathcal{J}(\lambda + \epsilon)).$$

Note that for $x \in \mathcal{J}(\lambda) \setminus \mathcal{J}(\lambda + \epsilon)$ we have

$$0 \leq \frac{1}{\gamma} - \lambda \frac{dQ}{dP}(x) < \epsilon \frac{dQ}{dP}(x).$$

This gives the following:

$$\epsilon \cdot Q(\mathcal{J}(\lambda + \epsilon)) \leq h(\lambda) - h(\lambda + \epsilon)$$
$$\leq \epsilon \cdot Q(\mathcal{J}(\lambda + \epsilon)) + \epsilon \cdot Q(\mathcal{J}(\lambda) \setminus \mathcal{J}(\lambda + \epsilon))$$
$$= \epsilon \cdot Q(\mathcal{J}(\lambda)),$$

which shows that $h$ is continuous. Also

$$\lim_{\epsilon \to 0^+} \frac{h(\lambda + \epsilon) - h(\lambda)}{\epsilon} = \lim_{\epsilon \to 0^+} -Q(\mathcal{J}(\lambda + \epsilon))$$
$$= -Q(\mathcal{J}(\lambda)).$$

It is obvious that $h(0) = 1/\gamma$ and $h \leq \gamma^{-1}$ for $\lambda \geq 0$. By Jensen's inequality applied to the convex function $x \mapsto (a - x)_+$, we have $h(\lambda) \geq \left( \gamma^{-1} - \lambda \right)_+$. So $h(1) \geq \gamma^{-1} - 1$. We conclude that $h$ may reach the value $(1 - \gamma)/\gamma = \gamma^{-1} - 1$ only on $[1, +\infty)$. Note that

$$h(\lambda) \to \frac{1}{\gamma} P \left( \frac{dQ}{dP}(X) = 0 \right) = \frac{\Delta}{\gamma} \geq 0 \quad \text{as} \quad \lambda \to \infty.$$

Thus if $\Delta/\gamma > \gamma^{-1} - 1$ the equation $h(\lambda) = \gamma^{-1} - 1$ has no solutions, as $h$ is non-increasing. If $\Delta/\gamma = \gamma^{-1} - 1$ then either $h(\lambda) > \gamma^{-1} - 1$ for all $\lambda \geq 0$ and we have no solutions or there is a finite $\lambda' \geq 1$ such that $h(\lambda') = \gamma^{-1} - 1$, which means that the equation is also satisfied by all $\lambda \geq \lambda'$, as $h$ is continuous and non-increasing. Finally, if $\Delta/\gamma < \gamma^{-1} - 1$ then there is a unique $\lambda^\dagger$ such that $h(\lambda^\dagger) = \gamma^{-1} - 1$, which follows from the convexity of $h$. ∎

Next we introduce some simple convenience lemma for comparing convex functions of random variables.

**Lemma 8** *Let $f$ be a convex function, $X, Y$ be real-valued random variables such that $X \leq Y$ a.s., and $c \in \mathcal{R}$ be a constant such that*[8]

$$\mathbb{E}\left[\min(c, Y)\right] = \mathbb{E}\left[X\right].$$

*Then we have the following lower bound:*

$$\mathbb{E}\left[f(X) - f(\min(c, Y))\right] \geq 0.$$

**Proof** We decompose the expectation with respect to the value of the min, and use the convexity of $f$:

$$
\begin{aligned}
f(X) &- f(\min(c, Y)) \\
&= \mathbf{1}_{[Y \leq c]}(f(X) - f(Y)) + \mathbf{1}_{[Y > c]}(f(X) - f(c)) \\
&\geq \mathbf{1}_{[Y \leq c]}f'(Y)(X - Y) + \mathbf{1}_{[Y > c]}(X - c)f'(c) \\
&\geq \mathbf{1}_{[Y \leq c]}f'(c)(X - Y) + \mathbf{1}_{[Y > c]}(X - c)f'(c) \\
&= Xf'(c) - \min(Y, c)f'(c),
\end{aligned}
$$

where we used the fact that $f'$ is non-decreasing in the previous to last step. Taking the expectation we get the result. ∎

**Lemma 9** *Let $P_g, P_d$ be two fixed distributions and $\beta \in (0, 1)$. Assume*

$$P_d\left(\frac{dP_g}{dP_d} = 0\right) < \beta.$$

*Let $\mathcal{M}(P_d, \beta)$ be the set of all probability distributions $T$ such that $(1 - \beta)dT \leq dP_d$. Then the following minimization problem:*

$$\min_{T \in \mathcal{M}(P_d, \beta)} D_f(T \| P_g)$$

*has the solution $T^*$ with density*

$$dT^* := \min(dP_d/(1 - \beta), \lambda^\dagger dP_g),$$

*where $\lambda^\dagger$ is the unique value in $[1, \infty)$ such that $\int dT^* = 1$.*

**Proof** We will use Lemma 8 with $X = dT(Z)/dP_g(Z)$, $Y = dP_d(Z)/\left((1 - \beta)dP_g(Z)\right)$, and $c = \lambda^*$, $Z \sim P_g$. We need to verify that assumptions of Lemma 8 are satisfied. Obviously, $Y \geq X$. We need to show that there is a constant $c$ such that

$$\int \min\left(c, \frac{dP_d}{(1 - \beta)dP_g}\right) dP_g = 1.$$

Rewriting this equation we get the following equivalent one:

$$
\begin{aligned}
\beta &= \int \left(dP_d - \min\left(c(1 - \beta)P_g, dP_d\right)\right) \\
&= (1 - \beta)\int \left(\frac{1}{1 - \beta} - c\frac{dP_g}{dP_d}\right)_+ dP_d.
\end{aligned}
\tag{19}
$$

Using the fact that

$$P_d \left( \frac{dP_g}{dP_d} = 0 \right) < \beta$$

we may apply Lemma 7 and conclude that there is a unique $c \in [1, \infty)$ satisfying (19), which we denote $\lambda^\dagger$. ∎

To conclude the proof of Theorem 2, observe that from Lemma 9, by making the change of variable $T = (P_d - \beta Q)/(1 - \beta)$ we can rewrite the minimization problem as follows:

$$\min_{Q: \, \beta dQ \leq dP_d} D_{f \circ} \left( P_g \, \| \, \frac{P_d - \beta Q}{1 - \beta} \right)$$

and we verify that the solution has the form $dQ_\beta^\dagger = \frac{1}{\beta} \left( dP_d - \lambda^\dagger (1 - \beta) dP_g \right)_+$. Since this solution does not depend on $f$, the fact that we optimized $D_{f \circ}$ is irrelevant and we get the same solution for $D_f$.

## G.4 Proof of Lemma 2

The first inequality follows from the optimality of $Q_\beta^*$ (hence the value of the objective at $Q_\beta^*$ is smaller than at $P_d$), and the fact that $D_f$ is convex in its first argument. The second inequality follows from the optimality of $Q_\beta^\dagger$ (hence the objective at $Q_\beta^\dagger$ is smaller than its value at $P_d$ which itself satisfies the condition $\beta dP_d \leq dP_d$). For the third inequality, we combine the second inequality with the first inequality of Lemma 1 (with $Q = R = Q_\beta^\dagger$).

## G.5 Proof of Corollaries 1 and 2

For Corrollay 1, combine Lemma 1, Theorem 1, and Lemma 2. Corollary 2 immediately follows from Lemma 1, Theorem 2, and Lemma 2. It is easy to verify that for $\gamma < \beta/4$, the coefficient is less than $(\beta/2 + \sqrt{1 - \beta})^2 < 1$ (for $\beta > 0$).