[Reviews · NeurIPS 2017]

Reviewer 1



This paper builds a mega-algorithm that can incorporate various sub-models to tackle with missing mode problem. It tactfully applied AdaBoost and other mixture model spirits in the context of GAN. The paper theoretically analyzed the optimal and suboptimal distributions that are added as mixture components, and get the corresponding convergence results. The paper is well organized and the details are clearly elaborated. There are some places that are better to be explained more clearly: 1. Whether the mixture model weight matches well the corresponding mode height? Many of the results(median) in table 1 is larger than 0.95 and it may suggest some mode drop. In the Toy Dataset, it can possibly be shown by changing 0.95 in calculating C to be 0.1 or even 0.05 and see the results. 2. In the calculation of data point weights, the algorithm requires an optimum discriminator D_M between the original data and the current mixture. Can this optimum discriminator be obtained during training in practice? If not, how does this influence the data points weight and the new model component? In general, the paper is theoretically sound and the results are supportive.

Reviewer 2



The paper proposes a new method inspired by AdaBoost to address the missing mode problem often occurs in GAN training. In general, the paper is quite well written. The studied problem is interesting and important, and the theoretical analyses seem solid. The proposed algorithm is novel and shows good empirical results on synthetic dataset. Below are my minor comments: 1. I know it is probably due to space limit, but it would be good if the authors can provide more explanation of the intuition on the theoretical results such as Theorem 1 and 2. This will make the paper much easier to understand. 2. The proposed method does not seem to have significant advantage over the standard GAN on real datasets such as MNIST and MNIST3. It would be good if the authors can try more datasets such as ImageNet. Otherwise the practical applicability is still in question.

Reviewer 3



AdaGAN is a meta-algorithm proposed for GAN. The key idea of AdaGAN is: at each step reweight the samples and fits a generative model on the reweighted samples. The final model is a weighted addition of the learned generative models. The main motivation is to reduce the mode-missing problem of GAN by reweighting samples at each step. It is claimed in the Introduction that AdaGAN can also use WGAN or mode-regularized GAN as base generators (line 55). This claim seems to be an overstatement. The key assumption of AdaGAN is that the base generator aims at minimizing the f-divergence (as mentioned in Line 136-137). This assumption does not hold for WGAN or mode-regularized GAN: WGAN minimizes the Wasserstein distance, and the mode-regularized GAN has an additional mode regularizer in the objective. WGAN and mode-regularized GAN are state-of-the-art generative models targeting the mode-missing problem. It is more convincing if the author can demonstrate that AdaGAN outperforms the two algorithms. Corollary 1 and 2 assume that the support of learned distribution P_g and the true distribution P_d should overlap for at least 1-beta. This is usually not true in practice. A common situation when training GAN is that the data manifold and the generation manifold are disjoint (see e.g., the WGAN paper).